# Population Pharmacokinetic Study of Benzylpenicillin in Critically Unwell Adults

**DOI:** 10.3390/antibiotics12040643

**Published:** 2023-03-24

**Authors:** Reya V. Shah, Karin Kipper, Emma H. Baker, Charlotte I. S. Barker, Isobel Oldfield, Barbara J. Philips, Atholl Johnston, Jeffrey Lipman, Andrew Rhodes, Marina Basarab, Mike Sharland, Sarraa Almahdi, Rachel M. Wake, Joseph F. Standing, Dagan O. Lonsdale

**Affiliations:** 1Institute for Infection and Immunity, St George’s, University of London, London SW17 0RE, UK; 2Department of Clinical Pharmacology & Therapeutics, St George’s University Hospitals NHS Foundation Trust, London SW17 0QT, UK; 3Institute of Chemistry, University of Tartu, 50411 Tartu, Estonia; 4Analytical Services International Ltd., London SW17 0RE, UK; 5Department of Medical and Molecular Genetics, King’s College London, London WC2R 2LS, UK; 6Brighton and Sussex Medical School, Brighton BN1 9PX, UK; 7Clinical Pharmacology, William Harvey Research Institute, Queen Mary University of London, London WC1E 7HU, UK; 8Jamieson Trauma Institute, Royal Brisbane and Women’s Hospital, Brisbane, QLD 4029, Australia; 9The University of Queensland Centre for Clinical Research, Brisbane, QLD 4029, Australia; 10Division of Anaesthesiology Critical Care Emergency and Pain Medicine, Nimes University Hospital, University of Montpellier, 30029 Nimes, France; 11Department of Critical Care, St George’s University Hospitals NHS Foundation Trust, London SW17 0QT, UK; 12Infection Care Group, St George’s University Hospitals NHS Foundation Trust, London SW17 0QT, UK; 13London North West University Healthcare NHS Trust, London HA1 3UJ, UK; 14Clinical Academic Group in Infection and Immunity, St George’s University Hospitals NHS Foundation Trust, London SW17 0QT, UK; 15Great Ormond Street Hospital for Children NHS Foundation Trust, London WC1N 3JH, UK; 16UCL Great Ormond Street Institute of Child Health, London WC1N 1EH, UK

**Keywords:** benzylpenicillin, pharmacokinetics, critical illness, beta-lactam, antibiotic, NONMEM

## Abstract

Pharmacokinetics are highly variable in critical illness, and suboptimal antibiotic exposure is associated with treatment failure. Benzylpenicillin is a commonly used beta-lactam antibiotic, and pharmacokinetic data of its use in critically ill adults are lacking. We performed a pharmacokinetic study of critically unwell patients receiving benzylpenicillin, using data from the ABDose study. Population pharmacokinetic modelling was undertaken using NONMEM version 7.5, and simulations using the final model were undertaken to optimize the pharmacokinetic profile. We included 77 samples from 12 participants. A two-compartment structural model provided the best fit, with allometric weight scaling for all parameters and a creatinine covariate effect on clearance. Simulations (n = 10,000) demonstrated that 25% of simulated patients receiving 2.4 g 4-hourly failed to achieve a conservative target of 50% of the dosing interval with free drug above the clinical breakpoint MIC (2 mg/L). Simulations demonstrated that target attainment was improved with continuous or extended dosing. To our knowledge, this study represents the first full population PK analysis of benzylpenicillin in critically ill adults.

## 1. Introduction

Infection in the critical care setting is a major cause of mortality and morbidity [1]. It is also common, with over 2/3 of critically unwell patients receiving antibiotics at some point during an admission to intensive care [2]. Early identification of an infection, prompt initiation of antibiotic therapy, and source control remain the key priorities in reducing mortality associated with sepsis [1,3]. Antibiotic dosing is often based on data from in vitro, animal, or healthy volunteer studies, or on studies in the non-critically ill [4]. However, pathophysiological changes associated with critical illness result in markedly altered pharmacokinetics (PK) and pharmacodynamics (PD) of drugs such as antibiotics [5,6]. Roberts et al. [4] demonstrated in an observational pharmacokinetic study of antibiotics that many critically ill patients do not achieve recognized therapeutic concentrations of these drugs and that suboptimal PK exposure of an antibiotic was associated with treatment failure. 

Benzylpenicillin is a beta-lactam antibiotic with primarily Gram-positive activity. It is an important agent in targeted therapy for bacterial infections such as pneumococcal pneumonia, invasive group A Streptococcus, and viridans group streptococci.

Beta-lactam antibiotics such as benzylpenicillin display time-dependent activity—bacterial killing and treatment efficacy positively correlate with the proportion of time that free drug concentration is above the minimum inhibitory concentration (%fT>MIC) of the organism being treated [4,7,8]. The optimal %fT>MIC for benzylpenicillin is unknown. Data from animal models have shown 90–100% survival of subjects infected with Streptococcus pneumoniae with 40% fT>MIC of benzylpenicillin, but data from observational studies in humans suggest higher fT>MIC should be targeted [9,10,11]. Data from other beta-lactams support longer (%fT>MIC) and higher (e.g., 2–5× MIC) exposures in critical illness, particularly for Gram-negative infections [12,13]. The recommended target for beta-lactams in critically unwell patients is 100% fT>MIC < 4× MIC [12,13].

The clinical breakpoint MIC, published by the European committee of antimicrobial susceptibility testing (EUCAST) [14], varies with species, type of infection, and dose of benzylpenicillin. The highest MIC at which benzylpenicillin might be considered effective is 2 mg/L (*Streptococcus pneumoniae* and viridans group streptococcus). A dose of 2.4 g 4-hourly is recommended for *Streptococcus pneumoniae* isolates with an MIC of 1–2 mg/L. This dose is also recommended in cases of endocarditis caused by *Streptococcus species* with an MIC of >0.125 mg/L (and ≤0.5 mg/L) [15]. The clinical breakpoint MIC for other Streptococcal species/infection sites are lower. For example, in Groups A, B, C, and G Streptococcus, the highest susceptible MIC is 0.25 mg/L for indications other than meningitis, but for meningitis with Group B Streptococcus (*Streptococcus agalactiae)*, the clinical breakpoint MIC is 0.125 mg/L.

Multiple studies have demonstrated the large pharmacokinetic variability of beta-lactam antibiotics in critically unwell patients [4,6,8], with many patients receiving suboptimal antimicrobial exposure when receiving standard intravenous bolus dosing regimens. Alternative dosing strategies, such as continuous or extended infusions, have been used for some beta-lactams [7,16,17]. These have been shown to improve drug exposure and target attainment [7,16,17]. To date, there are no published models for benzylpenicillin that investigate the effect of critical illness on its pharmacokinetics. There are two published studies of benzylpenicillin PK that address use in the non-critically unwell [18] and in endocarditis patients [19]. Given the importance of benzylpenicillin as a narrow spectrum agent for common serious streptococcal infections, improved understanding of its pharmacokinetics in critically unwell patients is essential to ensure optimal antimicrobial therapy.

We undertook an observational study of benzylpenicillin pharmacokinetics in critically ill adults to model pharmacokinetics and assess the probability of attaining the recognized pharmacokinetic–pharmacodynamic targets. Some of this data were used to inform a whole-life beta-lactam antibiotic model and have been published previously [8]. The study presented here focuses on benzylpenicillin pharmacokinetics in adults and includes a full covariate analysis and simulations for alternative dosing strategies. 

## 2. Results

Twelve patients received benzylpenicillin and contributed 80 plasma samples. Two patients received 2.4 g 4-hourly; two received 1.2 g 6-hourly; and eight received 1.2 g 4-hourly. A summary of patient characteristics is presented in Table 1, with the raw pharmacokinetic data presented in Figure 1. Concentrations from different dosing intervals are included in the data for each individual (represented by a line); therefore, concentrations may appear to be increased when they are from separate intervals. The median age was 57.7, with equal sex distribution. Female patients had a lower median age (52) and weight (71.5 kg) compared with those of male patients (age 62.8; weight 80 kg). Most patients were receiving antibiotics for a lower respiratory tract infection, including community-acquired pneumonia, hospital-acquired pneumonia, and aspiration pneumonia, or soft tissue infections. Eleven patients of the twelve were successfully discharged home, with one of those dying at home within the follow-up period, and one patient who was receiving treatment for infective endocarditis died in hospital. One patient was receiving renal replacement therapy (RRT) during the study period. For this participant, only the pharmacokinetic samples drawn following the recovery of renal function and cessation of RRT were included in the analysis. A total of 77 pharmacokinetic samples were included in the analysis.

The model build summary is shown in Appendix A. A two-compartment structural model was found to provide the optimal structural fit (reduction in OFV of 104.4 from one-compartment). Parameters representing interindividual variability were added to all parameters other than Q, for which this parameter was found to be negligible. 

Addition of creatinine to clearance (see Equation (1)) significantly improved the model fit (OFV reduced by 19.1). Further addition of covariates—height, BMI, sex, serum albumin, and temperature—did not improve the model fit. Parameter estimates for the final model are shown in Table 2. Goodness-of-fit plots (Figure 2) and visual predictive checks (Figure 3) of the final model are presented, demonstrating a good model fit. 

Equation (1): Typical value of clearance:


(1)
$$\text{CL}={\;\theta }_{\text{CL}}{\text{e}}^{\eta_{1}}{\left(\frac{\text{Creatinine}}{70}\right)}^{\theta_{\text{creat}}}$$


Simulations of four dosing strategies were performed to estimate %fT>MIC at various MIC values. Dosing strategies simulated included (a) standard dosing of 1.2 g bolus 4-hourly, (b) 2.4 g bolus 4-hourly, (c) 1.2 g bolus followed by 6 g continuous infusion over 24 h, (d) 7.2 g continuous infusion over 24 h, and (e) 1.2 g extended infusion over 2 h, 4-hourly. The results are displayed in Figure 4, with median values for %fT>MIC with each dosing regimen presented in Appendix A.

At a dose of 1.2 g 4-hourly, 38% of simulated patients failed to achieve the conservative target of 40% fT>MIC of 2 mg/L. This dose is unlikely to be recommended at such high MIC, although it may be chosen as an empirical dose prior to the knowledge of the organism and sensitivity. At a dose of 2.4 g 4-hourly at the same MIC, there were still 11% of simulated patients failing to achieve 40% fT>MIC. A higher target of 100% fT>MIC is recommended for critically unwell patients. At a dose of 1.2 g 4-hourly, 36% of simulated patients achieved this for an MIC of 1 mg/L, and the same proportion achieved this target with a dose of 2.4 g 4-hourly and an MIC of 2 mg/L. With an extended infusion of 1.2 g over 2 h, 4-hourly, all simulated patients achieved above 40% fT>MIC, and 35% of simulated patients achieved above 98% fT>MIC for an MIC of 2 mg/L. For the continuous infusion simulations (both with and without loading dose), 95% of simulated patients achieved concentrations above 2 mg/L for over 97% of the entire 24-h simulated period. 

## 3. Discussion

To our knowledge, this is the first study of benzylpenicillin PKPD in critically ill adults. We have extended upon the work in [8] through presenting a full covariate analysis and simulations of alternative dosing strategies. A two-compartment model was found to provide the best fit for the data, with creatinine as a covariate effect on clearance. Model evaluation methods suggest a robust fit of the model to the data.

A comparison of our findings to the two studies published in non-critically ill populations [18,19] demonstrate some differences. A population pharmacokinetic study performed by Bos et al. in a non-ICU adult population receiving intravenous benzylpenicillin in a hospital in Mozambique [18] found a one-compartment model, with creatinine clearance (CrCl) as a covariate effect on clearance, provided the best fit for their data. Primary pharmacokinetic parameter estimates from this study were 29 L/h for clearance and 40 L for volume of distribution. These are higher than the values estimated in our study—clearance was 23.1 L/h/70 kg, and volume of distribution was 24.9 L/70 kg. Of note, the median weight in the Mozambiquan study was 49 kg, whereas our parameters were scaled to 70 kg. The level of difference in pharmacokinetic parameters between our work and that of Bos et al. is, therefore, larger than the absolute difference in presented values due to this weight disparity. The difference in pharmacokinetic parameter estimates is most likely explained by the effect of critical illness and/or the demographic and physiological diversity between the two populations studied. 

Öbrink-Hansen et al. presented an analysis of PKPD target attainment in patients being treated for infective endocarditis with an antibiotic regimen including benzylpenicillin [19]. The dose used was 3 g 4-hourly, which was higher than in our study. In common with our study, the authors found marked interindividual variability, and at least 25% of patients did not achieve 100% fT>MIC, although the majority of patients achieved the PKPD target of 50% fT>MIC recommended for non-critically unwell patients. With only two PK samples per patient, the authors noted that they were unable to make a full PK profile for benzylpenicillin. A random coefficient model demonstrated an association between age, creatinine clearance, and albumin with benzylpenicillin concentration. 

We have also, similarly to the two studies described [18,19], found a creatinine effect on model clearance. However, unlike the two studies noted, we have chosen not to test CrCl as a covariate effect on clearance, as this was not directly measured during the study and estimates for CrCl are not validated in critically ill populations.

The benzylpenicillin elimination half-life estimated from our study was 67 min. This is higher than that estimated by Bos et al. (57 min) [18] and the half-life of approximately 30 min referenced in the benzylpenicillin summary of product characteristics (SPC) [20]. Lower clearance compared with those of the standard values has been noted for other antimicrobials studied in ABDose [8], and as previously described, this may represent a failure of clearance mechanisms as part of critical illness, which has not been adequately captured by the creatinine covariate effect on clearance in our model. Of note, there are three concentration profiles in Figure 1 that are markedly higher than those of other participants. All three participants had acute kidney injury, with creatinine ranging from 176–486, emphasizing that creatinine alone is an imperfect marker of renal injury.

The high residual standard error (%RSE) on intercompartmental clearance (Q) of 50% may be due to the small number of patients and may indicate that a similar fit might be obtained with a one-compartment model, similar to the findings of Bos et al. [18].

We found marked variability in parameter estimates between participants in this study, with greater than 10-fold differences in half-life and clearance between individuals, and approximately 3-fold differences in volume of distribution (Table 2). This finding of extreme pharmacokinetic variability is consistent with those of other studies measuring antimicrobial pharmacokinetics in critical illness [4,6,8]. For clearance in particular, it is likely that this range of results indicates the effect of acute kidney injury (low clearance) and, for some patients, the presence of augmented renal clearance of benzylpenicillin (high clearance) [21].

The simulations performed suggest poor PK/PD target attainment for critically unwell adults receiving the traditional bolus dosing of benzylpenicillin and infected with organisms of the EUCAST resistant breakpoint MIC of 2 mg/L for *Streptococcus pneumoniae* and viridans streptococci. A PK/PD target of 100%fT>MIC has been recommended for critically unwell patients. In our simulations, one in ten patients achieved less than 80% fT>MIC of 0.25 mg/L (the susceptible EUCAST breakpoint) at the standard recommended benzylpenicillin dose of 1.2 g 4-hourly, and one in ten patients achieved less than 80% fT>MIC of 0.5 mg/L at a dose of 2.4 g 4-hourly. The simulations suggested that PK/PD target attainment with extended infusions of 1.2 g infusions over 2 h at 4-hourly intervals are comparable to those of 2.4 g 4-hourly bolus dosing, with 6% of simulated patients achieving less than 80% fT>MIC of 0.5 mg/L.

Our simulations suggest that PK/PD target attainment is much improved with a continuous infusion of 7.2 g in 24 h, or a loading dose of 1.2 g followed by a continuous infusion of 6 g over 24 h. Over 95% of patients receiving these regimens would achieve 97–100% fT>MIC for MIC less than or equal to 2 mg/L, and 100% of patients would achieve 98–100% fT>MIC for MIC of 1 mg/L or below. Target attainment at higher MIC is improved with the use of a loading dose at the start of an infusion, with 82% of patients achieving 100% fT>MIC and 97% achieving 99% fT>MIC of 2 mg/L. Both of these infusion strategies use the same total daily dose as that of the 1.2 g 4-hourly bolus regimen. Continuous infusion strategies may provide a method to optimize antimicrobial exposure for organisms with higher MIC. This is likely to be of particular utility for critically unwell patients with augmented renal clearance [21]. Where vascular access is challenging, extended infusions may be preferred. 

This study has a number of limitations. The sample size of 12 is small, although multiple plasma samples were taken per patient, and this is not an unusual sample size in PK studies of this type. The measurement of pharmacodynamic endpoints was limited to discharge or death at follow-up, and since the study was not designed to examine for meaningful patient outcomes, we cannot draw conclusions between pharmacokinetic profiles and clinical outcome. Clinical outcome studies are still required to confirm benefit from improving benzylpenicillin PK/PD target attainment. A range of % protein binding is noted in the literature for benzylpenicillin, and we have chosen to use the SPC reference of 60% [20], which will impact upon the simulation results. 

From EUCAST’s collation of data of MIC distributions of wild-type organisms [22], 100% of Group A Streptococcus, 92.5% of Streptococcus pneumoniae, and 84.6% of viridans group streptococci studied had an MIC of less than or equal to 0.25 mg/L for benzylpenicillin. Our simulations demonstrated that 66% of patients receiving 1.2 g 4-hourly and 82% receiving 2.4 g 4-hourly would achieve the recommended PK/PD target of 100% fT>MIC with an MIC of 0.25 mg/L. 

EUCAST’s data show that only 2.4% of Streptococcus pneumoniae and 7.5% of viridans group Streptococci studied have an MIC above but not equal to 1 mg/L. For an MIC of 1 mg/L, our simulations found that 36% of patients simulated to receive 1.2 g 4-hourly and 49% of those receiving 2.4 g 4-hourly would achieve the recommended PK/PD target for these organisms with a higher MIC. Whilst organisms with higher MIC closer to the breakpoint of 2 mg/L are more likely to require alternative dosing strategies in critical illness to optimize pharmacokinetic profiles, the EUCAST data show that these organisms are less common amongst wild-type Streptococci. 

## 4. Materials and Methods

This benzylpenicillin pharmacokinetic study formed part of the ABDose study (Antibiotic Dosing), a multidrug, multi-age-group pharmacokinetic and pharmacodynamic study of antibiotics commonly used in critical illness [8]. 

Patients admitted to the intensive care unit (ICU) of St. George’s Hospital in London, United Kingdom and receiving benzylpenicillin were recruited. Informed consent was obtained from all subjects involved in the study either at enrollment or, in cases where critical illness temporarily impaired capacity, next of kin provided assent, and informed consent was obtained once the patient’s capacity was regained. Exclusion criteria were previous enrollment in ABDose, death expected within 48 h from enrollment, or treatment withdrawal for palliation. 

The study was conducted in accordance with the Declaration of Helsinki, and ethical approval was provided by the national research ethics (REC) committee London (REC reference 14/LO/1999). The study was sponsored by St. George’s University of London (Joint research office (JRO) reference 14.0195).

Demographic and clinical information were collected using the patients’ clinical notes. Baseline data were collected, including age, weight, height, comorbidities and indications for antimicrobial therapy, and admission to critical care. Evidence of organ dysfunction was gathered from routine observations, blood tests, and acute illness severity scoring. Microbiology results were recorded, including any positive cultures and MIC data. 

Antibiotic doses and timing were recorded from electronic prescriptions and drug delivery devices (infusion pumps) to ensure the accuracy of the drug administration data. Patients receiving benzylpenicillin were given either 1.2 g intravenously at 4-hourly or 6-hourly intervals or 2.4 g intravenously at 4-hourly intervals, with the dose determined by the treating clinicians. Pharmacokinetic samples were taken from radial arterial lines. An opportunistic sampling strategy was utilized to time with routine clinical samples, with an aim to obtain samples at specified time points within the dosing frame (Table 3). These were immediately placed upon ice and plasma separated by centrifugation. The plasma sample was then frozen for subsequent analysis. The measurements were performed by Analytical Services International Ltd., using tandem ultra-high-performance liquid chromatography-mass spectrometry. This method has previously been described [23]. 

The data for adults receiving benzylpenicillin during the ABDose study were analyzed for this work. Population pharmacokinetic analysis was performed using the nonlinear mixed effects modelling software NONMEM^®^ (version 7.5 ICON plc, Dublin, Ireland) [24], operating with GFortran (version 10.2.0). First-order conditional estimation method with interaction (FOCE-i) was used. R version 4.0.2 and R packages xpose4 [25,26] and Perl-speaks Nonmem (PsN version 5.3.0) [26,27] were used to produce goodness-of-fit plots and visual predictive curves. 

Weight was added to primary pharmacokinetic parameters a priori using allometric scaling. Compartment volumes were scaled with a fixed exponent of 1, whereas clearance parameters were scaled to an allometric exponent of 0.75, as previously described [8,28]. One- two-, and three-compartment models were tested to find the structural model best describing the data. 

Covariates were included in the model in a stepwise process. Choice of covariates was governed by clinical and physiological likelihood of the patient parameter having a significant effect: height, body mass index (BMI), sex, temperature, serum creatinine, serum albumin, and APACHE II score were tested. Model evaluation and selection was based on minimization of the NONMEM objective function value (OFV) (with nested models requiring a minimum reduction of 3.84 for a significant improvement in model fit at the level of *p* < 0.05 [24,29] for one additional parameter), review of parameter estimates for biological plausibility, goodness-of-fit plots, and assessment of model simulation properties. Non-parametric bootstrap analysis (n = 2000) was performed to calculate median and 95% confidence interval values using Perl-speaks Nonmem (PsN version 5.3.0) [27]. Elimination half-life was calculated using the final model parameter estimates. 

Simulations of 10,000 patients based on the final pharmacokinetic model were undertaken using the linpk package in R [30]. The following standard and alternative doses were simulated: (a) standard dosing of 1.2 g bolus 4-hourly, (b) 2.4 g bolus 4-hourly, (c) 1.2 g bolus followed by 6g continuous infusion over 24 h, (d) 7.2 g continuous infusion over 24 h, and € 1.2 g extended infusion over 2 h, 4-hourly. Plots were created to show the 95% prediction interval of fT above a range of MICs to estimate PKPD target attainment. Protein binding of 60% was assumed [20], with 40% remaining unbound and pharmacologically active.

## 5. Conclusions

We have presented the first full pharmacokinetic model of benzylpenicillin in critically ill adults. A two-compartment model with allometric scaling for weight and a creatinine covariate effect on clearance best describes benzylpenicillin pharmacokinetics in our study. We found marked variability in pharmacokinetic parameters between individuals, in keeping with those of pharmacokinetic studies of other drugs in critical illness. From our simulations, we demonstrated that PK/PD target attainment at standard doses of benzylpenicillin is likely to be adequate for the majority of infections in most patients, given the MIC of most wild-type organisms susceptible to benzylpenicillin. However, given the ongoing need to find strategies to reduce sepsis mortality and morbidity, alternative dosing regimens should be considered in order to optimize concentration profiles and provide cover for organisms with MIC up to the published clinical breakpoints. This is particularly important when using the drug empirically whilst awaiting culture and sensitivity results. Our work suggests continuous infusions would achieve this without an increase in the daily dose of benzylpenicillin. Clinical studies, alongside drug stability testing, are needed to validate these conclusions.

## Figures and Tables

**Figure 1 antibiotics-12-00643-f001:**
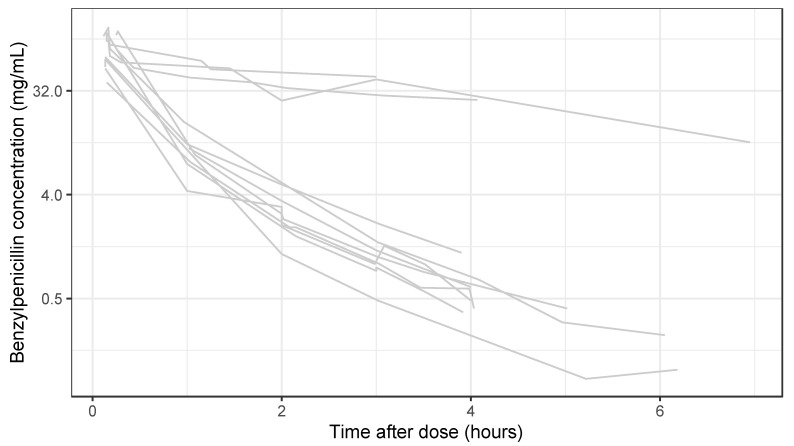
Measured benzylpenicillin concentrations. Each individual is represented by a separated line. Note that most individuals have data from multiple dosing intervals.

**Figure 2 antibiotics-12-00643-f002:**
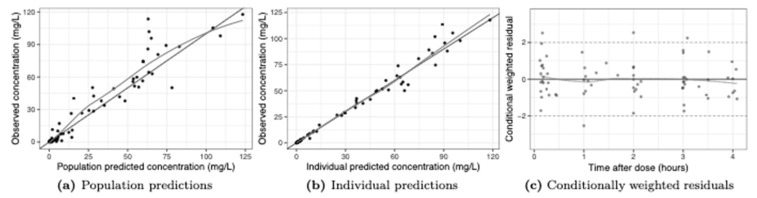
Goodness of fit plots for final benzylpenicillin model.

**Figure 3 antibiotics-12-00643-f003:**
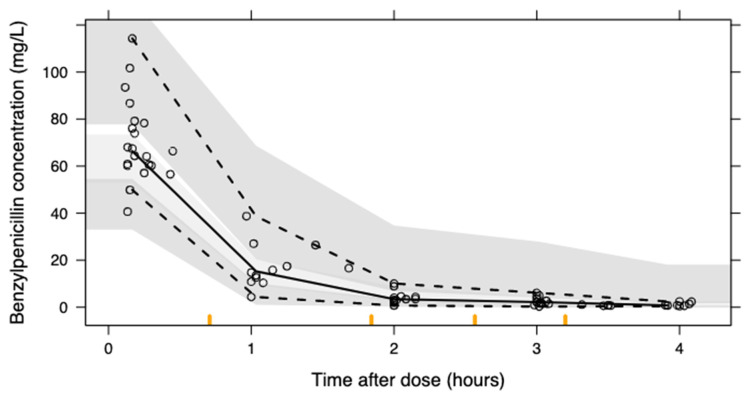
Final model prediction-corrected visual predictive check simulated to 4 hours. The prediction-corrected observed data is represented by the solid line, with the dashed lines representing the 95% confidence interval. The orange marks along the x-axis represent the binning of the time domain.

**Figure 4 antibiotics-12-00643-f004:**
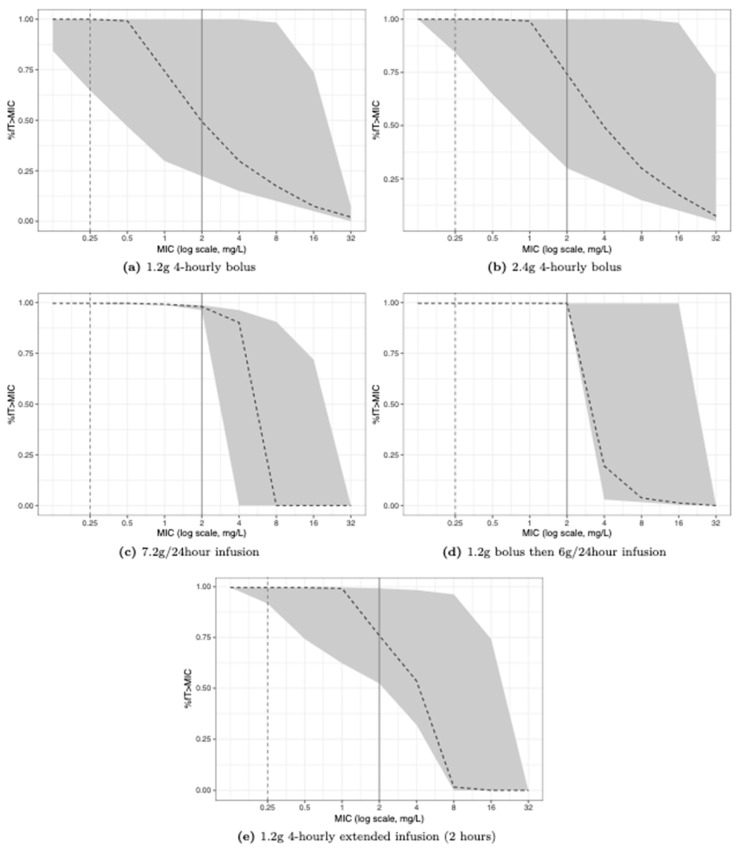
Time with free drug concentration above minimum-inhibitory concentrations (%fT>MIC) for simulated patients (n = 10,000) receiving a range of doses of benzylpenicillin. Median %fT>MIC indicated by dotted line. Shaded area represents 2.5th to 97.5th centiles (95% interval) of simulated patients. Vertical lines represent the resistant (solid) and sensitive (dashed) clinical breakpoint MIC.

**Table 1 antibiotics-12-00643-t001:** Summary of patient characteristics.

Characteristic	Median or n (Interquartile Range)	Full Range
No. of participants	12	-
Male: Female	6:6	-
Age	57.7 (44.3–63.2)	25.7–71.7
Height (cm)	172.0 (163.5–178.0)	150.0–188.0
Weight (kg)	70.0 (65.7–90.0)	60.0–120.0
BMI (kg/m^2^)	26.1 (22.1–27.9)	19.8–39.6
Ethnicity:		
Asian	2	
Caribbean	1	
White British	6	
White Irish	1	
Other/Not stated	2	
Infection source: *		
Soft tissue/Skin infection or abscess	5	
Lower respiratory tract infection	7	
Infective endocarditis	1	
Sepsis of unknown source	1	
Clinical outcome at 90-day review:		
Died in hospital	1	
Discharged home	10	
Died at home	1	
Serum creatinine (mmol/L)	70 (52–103.5)	34–486
Serum albumin (g/L)	28 (24–34)	14–43
CRP (mg/L)	133.9 (29.4–306)	13.6–386.5
APACHE II	14 (12.5–18)	5–23
Vasopressors (no. of patients)	5	
Ventilation (no. of patients with status recorded): **		
Intubated and ventilated	1	
Non-invasive ventilation	3	
Spontaneous ventilation	10	
Not recorded	1	

* Where more than one source is noted, both are recorded. ** Where ventilation status changed during the study, both statuses are recorded.

**Table 2 antibiotics-12-00643-t002:** Parameter estimates from final benzylpenicillin model.

	Mean Parameter Estimate (%RSE)	Individual Estimates (Range)	Bootstrap (n = 2000) Median (95% Interval)
Fixed effects			
θ_CL_ L/h/70 kg	23.1 (14)	4.1–53.1	24.0 (19.0–31.5)
θ_V1_ L/70 kg	15.1 (8)	9.9–27.6	14.4 (9.4–16.8)
θ_Q_ L/h/70 kg	11.1 (50)		11.4 (7.0–39.4)
θ_V2_ L/70 kg	9.8 (29)	6.5–19.4	10.5 (7.4–21.3)
θ_CREAT_	−0.916 (18)		−0.97 (−1.26–−0.67)
Random effects			
ω^2^_1_ CL (%CV)	42.0 (43)		40.0 (21.9–56.9)
ω^2^_2_ V1 (%CV)	22.6 (42)		22.8 (12.2–53.3)
ω^2^_3_ V2 (%CV)	20.5 (60)		20.2 (7.7–33.6)
Residual error			
σ^2^_1_ (proportional)	0.021 (47)		0.014 (0.006–0.032)
σ^2^_2_ (additive)	0.006 (67)		0.006 (0.003–0.031)
Derived parameters			
T_1/2_	1.11 h		

CL, clearance; V1, central volume of distribution; Q, intercompartmental clearance; V2, peripheral volume of distribution; θ_CREAT_, creatinine covariate effect (Equation (1)); T_1/2_, elimination half-life; %CV, coefficient of variation; %RSE, residual standard error.

**Table 3 antibiotics-12-00643-t003:** Dosing and scheduling sample.

Dosing Schedule	Sampling Interval 1	Sampling Interval 2
4-hourly	0.1, 2, 3.5, 4	0.2, 1, 3, 4
6-hourly	0.1, 2, 3, 6	0.1, 1, 5, 6

## Data Availability

The raw data is unfortunately unavailable as permission for open publication of pseudoanonymised data was not sought at time of consent of participants.

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
