# Peer review of "Population Pharmacokinetic Study of Benzylpenicillin in Critically Unwell Adults"

_antibiotics, 2023, doi:10.3390/antibiotics12040643_

Round 1
Reviewer 1 Report
This research is popPK modeling of benzylpenicillin in critically ill patients. Although only 12 subjects and 77 PK samples were used in this study, this study indicates important data. I have no comments. Results and Discussion are well written.
Formatting and typo errors need correction. It is also important to organize the format well not only the content when submitting a paper. There is a typo error @line 23-24 for example.
Author Response
Thank you for your time and effort in reviewing the manuscript. We are very grateful for your considered and helpful comments. We have corrected the typo at line 23-24 and reviewed the manuscript more generally for formatting, typos and corrections.
Reviewer 2 Report
Dear Authors,
Presented work is interesting and might be useful for clinical practice. However, it lacks some data.
Material and methods
What was normalised dose
Where nad how bolus was given . Was blood sample taken near given place of injection. How blood samples were prepared, how safe kept, how stored prior to analysis, how extracted, what was linearity, limit of detection limit of quantification.
What was age in male and female group. BMI.
What was time of sampling.
Results
Fig ' 1 there are three patients that did not have decline in concetratiin. Is there explanation.
Discussion
What was novelty compared to similar studies
Conclusion- should be more conected to results. No general discussion
Reference
Ref 30 is it complete
Ref 8 18 12 13 should be used in discussion in comparison to your study and what was novelty
Additionaly is clearance of creatinin used in model or not. You have two different information in the text.
Best of luck
Author Response
Thank you for your time and effort in reviewing the manuscript. We are very grateful for your considered and helpful comments. We have added additional responses for individual comments below:
Material and methods
What was normalised dose
The normalised dosing regimens given were 2.4 g 4-hourly, 1.2 g 4-hourly and 1.2 6-hourly. This has been clarified in the text.
Where nad how bolus was given . Was blood sample taken near given place of injection. How blood samples were prepared, how safe kept, how stored prior to analysis, how extracted, what was linearity, limit of detection limit of quantification.
Samples were drawn from indwelling vascular catheters. In this group, these were all radial arterial lines. There were no concentrations under the limit of detection. Details of the extraction and quantification method provided in reference [23].
The protocol for infusion was for 8 mins but exact times were taken from electronic infusion pumps and administration rates modelled using these.
What was age in male and female group. BMI.
We have added the difference in age and weight of the female and male group in the text. We've added BMI to Table 1.
What was time of sampling
Opportunistic sampling strategy to time with routine clinical samples with an attempt of getting samples at the times specified in new Table 3.
Results
Fig ' 1 there are three patients that did not have decline in concentration. Is there explanation.
None of the concentrations failed to decrease within a single dosing interval: some may look flat but (a). concentrations within multiple dosing intervals may be included within each individual’s raw data line; and (b). we’d point out the y-axis is logarithmic which may give an impression of failure to clear. There is a comment on renal injury.
What was novelty compared to similar studies
To our knowledge this is the first study of benzylpenicillin PKPD in critically ill adults with a full covariate analysis.
Conclusion
Should be more conected to results. No general discussion
We’ve adjusted the conclusion in light of your helpful comments. We hope the changes are acceptable.
Reference
Ref 30 is it complete
We have updated the reference information, many thanks.
Ref 8 18 12 13 should be used in discussion in comparison to your study and what was novelty
We have added some further comparison of our findings to other papers in the discussion. Ref 8 is work from this study that has been expanded upon in this paper. Ref 12 is a position paper from which we have taken the recommended PKPD targets quoted in this paper. Ref 13 supports the PKPD targets used in this paper. Ref 18 and 19 are studies of benzylpenicillin with discussion about direct comparisons.
Additionaly is clearance of creatinin used in model or not. You have two different information in the text.
Creatinine is used in the model, but creatinine clearance is not. We have adjusted the manuscript so that this is hopefully clearer.
Best of luck
Many thanks
Reviewer 3 Report
This paper reported Population pharmacokinetic study of benzylpenicillin in critically unwell adults. The manuscript did achieve a high enough priority score as it has the novelty, although there are two published studies of benzylpenicillin PK which address use in the non-critically unwell and inendocarditis patients. They reported 77 samples from 12 participants were included. A 2-compartment structural model provided the best fit, and simulations demonstrated that target attainment is improved with continuous or extended dosing. And, there were some flaws or uncertainties in this research in terms of my knowledge, so it should be revised before publish in my opinion.
1. As reported by authors, benzylpenicillin PK in the inendocarditis patients was reported, authors should compare this research as Ref.18.
2. 77 samples from 12 participants were too small, the authors should added the amounts.
3. How continuous or extended dosing should be elaborated in the paper?
Author Response
Thank you very much for your time and effort in reviewing the manuscript. We are very grateful for your considered and helpful comments.
- As reported by authors, benzylpenicillin PK in the in endocarditis patients was reported, authors should compare this research as Ref.18.
Thank you for this helpful comment. We have added a paragraph of comparison to the discussion.
- 77 samples from 12 participants were too small, the authors should added the amounts.
We recognise the small sample size but this is comparable to other studies in this patient group. We have noted this in our discussion as a limitation and the need for further work in this area.
- How continuous or extended dosing should be elaborated in the paper?
Apologies, we are not quite clear of the intended meaning of this comment. We have made some changes to the discussion which we hope the reviewer is satisfied with.
Round 2
Reviewer 2 Report
Dear Authors,
work is much improved.
Figures should be improved they are blurred.
Also I guess you did not understand question about sampling place. The question was did the samples were taken from same place where dose of antibiotic was administrated.
Normalised dose is dose ajusted devided to weight of patients. But your correction ia fine.
Also why mediana is presented not average value (i guess there is no normal distribution in any of data)
Good luck
Author Response
Thank you for your second round of comments.
We uploaded PDF versions of all the figures with our manuscript. These should be vector scaleable. Im not sure why they are blurred. Could the copy editors please comment.
With regards to sampling - we indicated in our response that the samples were from arterial lines. We thought it would be implicit that this is not the place where the drug was administered. If we need to be more specific we can be but our view is it is OK as is.
Thank you
Reviewer 3 Report
accept
Author Response
I have reviewed the manuscript for spelling. I cannot see any obvious spelling errors - there are some anglicised spellings (analysed for example). We will leave this to the copy editor to select style in this regard.
If there are other issues please could I ask for more specifics.
Thank you again for your time reviewing.